# Usability and agreement of the SWIFT-ActiveScreener systematic review support tool: Preliminary evaluation for use in clinical research

**Jenny J. W. Liu** [1,2]*, **Natalie Ein** [1,2], **Julia Gervasio** [1,3], **Bethany Easterbrook** [1,4], **Maede S. Nouri** [1], **Anthony Nazarov** [1,2,5], **J. Don Richardson** [1,2,6]

1 The MacDonald Franklin Operational Stress Injury Research and Innovation Centre, Lawson Health Research Institute, London, ON, Canada, 2 Department of Psychiatry, Schulich School of Medicine & Dentistry, Western University, London, ON, Canada, 3 Department of Psychology, Toronto Metropolitan University, Toronto, Canada, 4 Department of Psychology, Neuroscience & Behaviour, McMaster University, Hamilton, Canada, 5 Department of Psychiatry and Behavioural Neurosciences, McMaster University, Hamilton, Canada, 6 St. Joseph Operational Stress Injury Clinic, St. Joseph's Healthcare London, London, Ontario, Canada

* jenny.liu@sjhc.london.on.ca

## Abstract

Systematic reviews (SRs) employ standardized methodological processes for synthesizing empirical evidence to answer specific research questions. These processes include rigorous screening phases to determine eligibility of articles against strict inclusion and exclusion criteria. Despite these processes, SRs are a significant undertaking, and this type of research often necessitates extensive human resource requirements, especially when the scope of the review is large. Given the substantial resources and time commitment required, we investigated a way in which the screening process might be accelerated while maintaining high fidelity and adherence to SR processes. More recently, researchers have turned to artificial intelligence-based (AI) software to expedite the screening process. This paper evaluated the agreement and usability of a novel machine learning program, Sciome SWIFT-ActiveScreener (ActiveScreener), in a large SR of mental health outcomes following treatment for PTSD. ActiveScreener exceeded the expected 95% agreement of the program with screeners to predict inclusion or exclusion of relevant articles at the title/abstract assessment phase of the review and was reported to be user friendly by both novice and seasoned screeners. ActiveScreener, when used appropriately, may be a useful tool when performing SR in a clinical context.

## Introduction

Systematic reviews (SRs) are the current standard to collate and synthesize empirical evidence and evaluate trends across a specific body of literature in response to research questions. SRs involve strict structured and formal methodological processes [1, 2]. Standardized protocols,

**Data Availability Statement:** All relevant data are within the paper and its Supporting Information files.

**Funding:** This work was supported in part by the Atlas Institute for Veterans and Families. The funders had no role in study design, data collection and analysis, decision to publish, or preparation of the manuscript.

**Competing interests:** The authors have declared that no competing interests exist.

such as the Preferred Reporting Items for Systematic Reviews and Meta-Analysis (PRISMA), offer researchers a guideline for transparent and comprehensive reporting of SR processes and results [3]. In addition, formal guides established by Cochrane provide further evaluation criteria in order to provide appropriate context for the interpretation of study data in various research settings [4]. Despite these protocols, SRs continue to be a huge undertaking due to extensive resource requirements. Depending on the scope of review and precision of search terms used, researchers may review tens of thousands of articles during various stages of screening. Therefore, given the substantial resource and time commitment required to complete the screening phases for SRs, it is crucial to investigate opportunities which may accelerate the screening process.

The screening phases of a SR include de-duplicating search outputs across multiple databases, and screening title and abstract and full text [3]. During these steps, researchers examine each article against strict inclusion and exclusion criteria to determine its eligibility for inclusion in the SR. To ensure standards of quality, more than one individual must screen the same article independently at each screening stage, with the reliability between screeners calculated and reported as part of the standard requirements for publishing SR [5]. Altogether, screening phases can take hundreds of hours for each individual reviewer involved.

Artificial intelligence-based (AI) software such as COVIDENCE [6], CUREDATIS [7], and Sciome SWIFT-ActiveScreener [8] have been developed to help expedite SR screening and reduce the number of person-hours required to complete SRs. For the purpose of this paper, AI programs refer to programs that are enabled to perform tasks that normally require human intelligence during in the context of conducting a SR. While they do not eliminate human involvement in the screening process, each program may reduce time and resources spent using various proprietary solutions. For example, COVIDENCE aids clinical research reviews with its ability to distinguish between randomized controlled trials (RCTs) versus non-RCTs. Other SR tools include similar options to apply sorting tags and take notes. Critically, ActiveScreener, which is part of a growing set of novel tools harnessing active learning, can estimate completeness of the screening process, and notify reviewers when they may stop screening early. In this paper, we evaluated ActiveScreener in terms of its agreement with human screeners, and usability in a large SR of mental health outcomes following treatment for PTSD. ActiveScreener was selected for this project namely for its departure from programs that use AI to identify records, and instead, use machine learning to build a predictive algorithm to reduce time spent in screening phases of SRs.

ActiveScreener is a novel machine learning and web-based AI software for SRs. ActiveScreener uses a L2-regularized log-linear model and active-learning approach to screening, meaning that the model is continually trained during the screening process [8]. In this case, the "training" occurs whenever a screener identifies included articles through the real time manual identification of uploaded documents. Each time an article is identified as "included", the model is re-trained and re-orders the most likely relevant references to be screened next. A statistical model based on a negative binomial distribution is then used by ActiveScreener to estimate sensitivity of the screening process and is used to alert screeners that a specified threshold of likely relevant articles (in this case, 95%) have been included [8]. This AI prioritization of articles believed to be most relevant during screening can trim screening time and human effort by nearly 70% [8]. Past research utilizing ActiveScreener have found that the algorithms to work well in reviews involving the physical health literature [9, 10]. Despite indications of past use in health reviews, there is little evidence for how ActiveScreener may perform in evaluations of mental health and treatment outcomes. Further, the precision of the estimation model remains unclear. In this paper, we set out to evaluate the precision and usability of ActiveScreener in conducting screening for a mental health treatment SR [11].

Specifically, in Part 1, we formally evaluated the agreement between its predictive model relative to the screening outcomes conducted by individual human screeners, and in Part 2, we collected informal feedback regarding the usability of ActiveScreener amongst a cohort of screeners.

## Methods

### Participants

Eighteen screeners were trained to identify articles for inclusion and exclusion at the title and abstract assessment phase of the review and on the use of ActiveScreener for a meta-analysis and SR (for more details on this project see Liu et al., 2021). All respondents were paid employees or unpaid volunteers of the MacDonald Franklin Operational Stress Injury Research and Innovation Centre (MFOSIRC).

### Procedure

All respondents received an email with the link directing them to the online survey. Respondents were asked to complete both the demographic information and the ActiveScreener User Experiences Survey online via Google Forms. Data was collected in April 2022.

A total of 10,002 references required review at the title and abstract screening stage of this SR. ActiveScreener inclusion statistics were set at 95% predicted inclusion rate, resulting in 5,390 of these references to be reviewed in duplicate by screeners. Screeners were able to access ActiveScreener at any time on their own schedules, and when they logged on, they were provided with the most relevant article at that time as identified by ActiveScreener, rather than every screener reviewing articles in the same order. Once screening reached 95% of relevant articles included, according to ActiveScreener, all screeners stopped. At this stage, data consisting of the screening results for the 5,390 references reviewed by screeners and the remaining 4,612 references reviewed by the ActiveScreener AI were exported. Inclusion statistics were then reset to 100%, prompting the screeners to continue screening the remining 4,612 references for relevant abstracts to be screened in full-text. Data was once again exported. Screening results for the remaining 4,612 references from ActiveScreener and the screeners were then compared to assess agreement between ActiveScreener and reviewer's decisions during title and abstract screening.

No direct compensation was given for participating in this study, however; many of the respondents were paid employees of the MFOSIRC and completed the survey during working hours, thereby receiving nominal monetary compensation for the time spent participating. For unpaid volunteers, the time spent completing this survey was included in their volunteer hours, for which they are provided a letter of recognition.

### Measures

**Demographics.** Demographic information included: (1) the respondents' role within MFOSIRC, (2) whether the respondents conducted or assisted on a SR or meta-analysis prior to their time at the MFOSIRC, (3) respondents' level of experience with SR or meta-analyses (e.g., beginner, intermediate), and (4) types of software previously used by respondents for screening for SR or meta-analyses.

**ActiveScreener user experience survey.** This survey was created by authors (J.J.W.L & A. N) to capture respondents experiences using ActiveScreener. The survey consisted of 12 items (statements or questions) related to usability of ActiveScreener for screening (e.g., "SWIFT Active Screener is easy to use", "SWIFT Active Screener software was easy to learn"). Nine

statements were quantitative, and three questions were qualitative, providing open text boxes for responses. Of the quantitative items, eight statements were rated on a 5-point Likert scale, ranging from 'strongly agree' to 'strongly disagree', and one question was rated on a 5-point Likert scale, ranging from 'very confident' to 'not at all confident'. The qualitative items included three open-ended questions capturing information related to features of ActiveScreener the respondents enjoyed, any challenges experienced while using ActiveScreener, and any suggestions the respondents had to improve ActiveScreener. This survey was assessed for face validity, but as it was an internal assessment of usability and acceptability, no other reliability or validity assessments were undertaken.

### Data analysis

A confusion matrix and statistics were generated and used to evaluate the predictive agreement of ActiveScreener across three classes. The three classes were Included (represents references identified as meeting inclusion criteria), Excluded (representing references identified as meeting exclusion criteria), and Conflicted (representing disagreement on whether the reference should be included or excluded). Analyses were performed in R-Studio using the tidyverse [12], stringr [13], and caret [14] packages. Results are reported for only the title and abstract screening stage.

Both quantitative and qualitative data was used to provide descriptive information related to the respondents' experiences using ActiveScreener. For qualitative data, common themes were extracted from responses provided regarding enjoyable features of the software, challenges with ActiveScreener, and suggested improvements.

### Results

The multiclass confusion matrix for 4,612 references is presented in Table 1. As shown, both the screeners and the ActiveScreener AI identified 1,365 included references, 2,528 excluded references, and 622 conflicted references. For 97 references, the screeners identified these references as included, while the ActiveScreener AI identified these references as conflicted.

Overall, agreement was 97.9%, 95% Confidence Interval (CI) [0.97, 0.98], $p < .001$. Interrater reliability was reported with Kappa [Fleiss and Conger; 0.96]). Sensitivity for the three classes were: Included (0.93), Excluded (1.00), and Conflicted (1.00). Specificity for the three classes were: Included (1.00), Excluded (1.00), and Conflicted (0.98).

### Quantitative data

All 18 respondents completed all nine quantitative items. All respondents (100%) either agreed or strongly agreed that: their training needs were met; ActiveScreener was easy to learn; they felt confident using ActiveScreener; and they would recommend ActiveScreener for use in other SR. Nearly all respondents reported either agreeing or strongly agreeing that:

**Table 1. Confusion matrix (n = 4,612).**

| Predicted | | Actual | | |
|---|---|---|---|---|
| | | Conflicted | Excluded | Included |
| | Conflicted | 622 | 0 | 97 |
| | Excluded | 0 | 2528 | 0 |
| | Included | 0 | 0 | 1365 |

Notes. Actual = screeners; Predicted = ActiveScreener

ActiveScreener was easy to use (94.4%); and ActiveScreener had a user-friendly interface (94.5%). The majority of respondents (88.9%) also reported that they either agreed or strongly agreed that ActiveScreener had all of the features needed for adequate screening. Of the eight respondents who had prior experience with other screening programs or tools, seven respondents (87.5%) rated that they either agreed or strongly agreed that they preferred ActiveScreener over other programs. With regards to the experience of technical or system-related glitches, respondents varied in their perspectives, with 44.5% of respondents indicating that they experienced no technical or system-related glitches (either agreed or strongly agreed), while 22.2% indicated experiencing technical or system-related glitches (disagreed). Results for each survey items are reported in Table 2.

## Qualitative data

**Features enjoyed.** For the question capturing the features of ActiveScreener enjoyed most by respondents, three primary themes emerged from the data (see Table 3 for quotes).

*AI Predictability*. Respondents noted that ActiveScreener accelerates the screening process through predictive capabilities. Specifically, ActiveScreener reorders references based on individual patterns of inclusion and exclusion such that likely included articles are pushed to the top of the screening list.

*Screening process*. Respondents noted that ActiveScreener makes the screening process easier and faster. Specifically, all the information required for screening is available on one page including the article title, abstract, full text, and inclusion and exclusion criteria. This allows the screener to evaluate the article quickly.

*User-friendly interface*. Respondents noted that ActiveScreener has a user-friendly interface. For example, respondents noted ease of use and ability to access ActiveScreener from any device as a positive feature of this software.

**Challenges.** For the question capturing any challenges experienced by respondents, two primary themes emerged from the data (see Table 3 for quotes).

*Technical issues*. Respondents noted that they encountered some technical difficulties and glitches while using ActiveScreener. For example, connection loss specific to the ActiveScreener website or processing or loading speeds were commonly described.

*Article uploading*. Respondents noted that uploading articles individually to each reference is time consuming and could result in errors such as a mismatch of articles to references.

**Suggested improvements.** For the question capturing suggested improvements or additions to the program, three primary themes emerged from the data (see Table 3 for quotes).

*Data extraction*. Respondents noted that they would have liked the ability to either extract data directly within ActiveScreener or be able to export the included references with attached articles to other formats (e.g., SmartSheets).

*Bulk upload*. Respondents noted that they would like the ability to upload articles to references in bulk as opposed to one at a time.

*Interface improvements*. Respondents noted potential improvements to the user interface. For example, navigation opportunities, keeping a session counter of screened articles, and ability to flag references with incorrect articles attached.

## Discussion

In our study, we found that ActiveScreener performed above its expected 95% agreement in prediction at the title and abstract assessment phase of the SR and was found to be user friendly by both novice and seasoned screeners. Consistent with past evidence that the

**Table 2. Respondent data across measures (N = 18).**

| | n | % |
|---|---|---|
| **Demographic Information** | | |
| What is your role within MacDonald Franklin OSI Research Centre? | | |
| Project Lead/Co-Lead | 5 | 27.8 |
| Volunteer | 8 | 44.4 |
| Research Assistant (paid, full time) | 3 | 16.7 |
| Research Assistant (paid, part time) | 1 | 5.6 |
| Research Associate | 1 | 5.6 |
| Have you conducted/assisted in a systematic review/meta-analysis prior to your placement with us? | | |
| Yes | 9 | 50.0 |
| No | 9 | 50.0 |
| Level of experience in systematic review/meta-analyses. | | |
| Beginner (assisted in 3 or less) | 12 | 66.7 |
| Intermediate (lead one or engaged in 5 or less) | 3 | 16.7 |
| Advanced (lead multiple/engaged in 5 or more) | 3 | 16.7 |
| What softwares have you used for strictly screening in reviews?[a] | | |
| Swift ActiveScreener | 18 | 100 |
| Microsoft Excel (offline, via 365, or as google doc) | 7 | 33.9 |
| Smartsheets | 11 | 61.1 |
| Covidence | 6 | 33.3 |
| SysREV | 1 | 5.6 |
| EPPI-Reviewer | 0 | 0 |
| Distiller SR | 1 | 5.6 |
| SUMARI | 0 | 0 |
| Reference Management Softwares (e.g., Mendeley, Endnotes, etc.) | 0 | 0 |
| Other | 1 | 5.6 |
| **ActiveScreener User Experience Survey (Quantitative Items Only)** | | |
| SWIFT ActiveScreener is easy to use. | | |
| Strongly Agree | 6 | 33.3 |
| Agree | 11 | 61.1 |
| Neutral | 1 | 5.6 |
| Disagree | 0 | 0.0 |
| Strongly Disagree | 0 | 0.0 |
| Training to use the SWIFT ActiveScreener met my needs. | | |
| Strongly Agree | 11 | 61.1 |
| Agree | 7 | 38.9 |
| Neutral | 0 | 0.0 |
| Disagree | 0 | 0.0 |
| Strongly Disagree | 0 | 0.0 |
| SWIFT ActiveScreener was easy to learn. | | |
| Strongly Agree | 13 | 72.2 |
| Agree | 5 | 27.8 |
| Neutral | 0 | 0.0 |
| Disagree | 0 | 0.0 |
| Strongly Disagree | 0 | 0.0 |
| SWIFT ActiveScreener has all the features I need for screening. | | |
| Strongly Agree | 7 | 38.9 |
| Agree | 9 | 50.0 |

(*Continued*)

**Table 2.** (Continued)

| | n | % |
|---|---|---|
| **Demographic Information** | | |
| Neutral | 2 | 11.1 |
| Disagree | 0 | 0.0 |
| Strongly Disagree | 0 | 0.0 |
| SWIFT ActiveScreener is user friendly. | | |
| Strongly Agree | 5 | 27.8 |
| Agree | 12 | 66.7 |
| Neutral | 1 | 5.6 |
| Disagree | 0 | 0.0 |
| Strongly Disagree | 0 | 0.0 |
| SWIFT ActiveScreener does not have any technical/system glitches. | | |
| Strongly Agree | 1 | 5.6 |
| Agree | 7 | 38.9 |
| Neutral | 6 | 33.3 |
| Disagree | 4 | 22.2 |
| Strongly Disagree | 0 | 0.0 |
| I would recommend SWIFT ActiveScreener for use in screening with other reviews. | | |
| Strongly Agree | 10 | 55.6 |
| Agree | 8 | 44.4 |
| Neutral | 0 | 0.0 |
| Disagree | 0 | 0.0 |
| Strongly Disagree | 0 | 0.0 |
| I prefer SWIFT ActiveScreener over other platforms/softwares for screening. | | |
| Strongly Agree | 3 | 16.7 |
| Agree | 4 | 22.2 |
| Neutral | 1 | 5.6 |
| Disagree | 0 | 0.0 |
| Strongly Disagree | 0 | 0.0 |
| Not Applicable (have used no other software/platforms) | 10 | 55.6 |
| If you were to conduct another systematic review, how confident are you that you would use SWIFT ActiveScreener for citation screening? | | |
| Very confident—will absolutely use ActiveScreener | 8 | 44.4 |
| Confident—most likely will use ActiveScreener | 10 | 55.6 |
| Neutral–no preference | 0 | 0.0 |
| Not Confident–may use other software | 0 | 0.0 |
| Not at all Confident–will definitely use other software | 0 | 0.0 |

*Notes.* [a] indicates respondents could choose more than one answer.

effectiveness of this program can reduce screening time and effort by nearly 50% [15], we observed similar results with a large-scale review of PTSD treatment outcomes.

Regarding its agreement, our confusion matrix results indicated that when testing against a large-scale SR that included over 10,000 articles screened in the title and abstract phase, Active-Screener performed better than expected in its predictive algorithm. While the software was expected to reach 95% sensitivity, the actual agreement between the machine learning model and our screeners in this review exceeded 95% (97.9%), which may have been aided by the high number of independent screeners on this project. Further, of the categories examined,

**Table 3. ActiveScreener user experience survey qualitative feedback.**

| Survey Questions | Themes Identified | Examples of Respondent Quotes |
|---|---|---|
| What was the ActiveScreener feature you enjoyed the most? | AI Predictability | "It reorders studies based on screening patterns." |
| | Screening Process | "Having all the information on one page (title/abstract/full text) to decide whether to include or exclude. Love highlighting keywords." |
| | User-friendly Interface | "Simplicity of user interface." |
| What are some of the challenges you experienced with ActiveScreener? | Technical Issues | "Random software glitches where we had to reach out to the ActiveScreener team to find out what was happening." |
| | Article Uploading | "Uploading full text articles to the individual record." |
| What are some features you wish ActiveScreener would improve or add? | Data Extraction | "Making data extraction possible or easy to transfer all data to smartsheets with articles attached." |
| | Bulk Upload | "Bulk upload." |
| | Interface Improvements | "Being able to skip an abstract for the duration of a session (e.g., when a paper was attached to an incorrect abstract, I would skip it and go to the next abstract—but upon completion of the next abstract, the incorrectly-matched one would be next in queue.) Would be nice to be able to skip/flag/set aside without having to navigate away from it repeatedly." |

discrepancies between the predictive algorithm and actual human screening outcomes were minimal. Specifically, there were no discrepancies between human screeners and the Active-Screener AI with respect to articles that should be excluded from the SR. Only a small number of discrepancies were found between human screeners that indicated articles should be included, while the ActiveScreener AI predicted that the articles would be conflicted (i.e., predicted multiple human screeners would disagree on inclusion and exclusion) based on prior trends in human screening. This means that no studies that the ActiveScreener AI predicted to be included resulted in exclusions by screeners. Thus, these statistics, as yielded by the confusion matrix, indicate that ActiveScreener is a reliable and rigorous platform to accelerate screening at the title and abstract phase of SRs, especially when utilizing its predictive algorithm function. Future directions of this research should consider the assessment of Active-Screener AI agreement when including a fewer number of screeners and those with different levels of experience. Previous research in other areas indicates that ActiveScreener maintains high levels of agreement with as few as two reviewers [9], and duplication of this result would be beneficial for use in mental health-related SR with more limited resources. Further, to reduce human resources during screening, ActiveScreener should consider implementing new features such as bulk upload and templates for subsequent data extraction directly within the platform. Both would reduce the need for switching between programs when conducting reviews and would thereby reduce human resource requirements and the potential for error. Importantly, as decisions at the title and abstract phase were not compared directly against final inclusion decisions in this analysis, the magnitude of impact ActiveScreener has on the screener process in its entirety is not clear. However, one can assume with relative confidence that due to the high agreement in phase one, high levels of agreement would have been maintained in the final phase of full-text screening.

In examining user feedback amongst a group of screeners, we found that ActiveScreener was endorsed as easy to learn and easy to use. However, user feedback also noted that there were software glitches, such as the platform being unavailable from time to time, as well as glitches when uploading articles and using other features. While these challenges do not undermine its use, they provide areas of opportunity for ActiveScreener programmers to consider for future research and development. Of note, while the administered survey was developed internally and assessed for face validity, reliability and other forms of validity were not

examined. As such, this may have led to measurement error and quantitative results should be interpreted with caution.

## Conclusion

In considering the merits of ActiveScreener, it should be noted that the software's machine learning algorithm is reliant on the rigour of training and the strength of screeners that it bases its user feedback on. As such, users must conduct training and screening with care. In particular, the clarity in which inclusion and exclusion criteria may be applied during the initial screening stages is of vital importance in building the accuracy and agreement of the predictive model as well as for increasing agreement between human screeners and the model. Thus, researchers are encouraged to spend considerable time to ensure the inclusion and exclusion criteria are clearly understood and reliably applied by all screeners during the project training stages. In addition, another time-saving feature of ActiveScreener, the deduplication function for uploading references, can benefit from further development as it currently limits the deduplication to texts only, and does not extend to cover punctuation. Depending on the database, references may be exported with variable punctuations, which is not covered by the feature, resulting in many duplicate references when screening. However, it should be noted that this can easily be solved with workarounds, such as manually combining search yields on $r$ with generated codes that deduplicates references prior to uploading on ActiveScreener. Finally, it is important to note that ActiveScreener's program to accelerate the screening stage is only currently relevant at the title and abstract stage and excludes further reviews of full texts. Thus, current study findings and the potential time and resource savings are only applicable to the initial screening phase of SRs. Additionally, this paper did not present screening decisions at the title and abstract assessment phase relevant to the final sample of the included articles, and therefore can only describe how well ActiveScreener software performed compared to trained human screeners at this stage of the review process. Taken together, ActiveScreener appears to be a user friendly and valuable platform for SRs, and when used appropriately, may be a useful tool during the initial screening process.

## Author Contributions

**Conceptualization:** Jenny J. W. Liu, Bethany Easterbrook, Anthony Nazarov.

**Data curation:** Jenny J. W. Liu, Bethany Easterbrook.

**Formal analysis:** Jenny J. W. Liu, Natalie Ein, Julia Gervasio, Maede S. Nouri.

**Funding acquisition:** Anthony Nazarov, J. Don Richardson.

**Investigation:** Jenny J. W. Liu, Anthony Nazarov, J. Don Richardson.

**Methodology:** Jenny J. W. Liu, Anthony Nazarov.

**Project administration:** Jenny J. W. Liu, Bethany Easterbrook, Anthony Nazarov, J. Don Richardson.

**Resources:** Anthony Nazarov, J. Don Richardson.

**Software:** Anthony Nazarov.

**Supervision:** Jenny J. W. Liu, Anthony Nazarov, J. Don Richardson.

**Writing – original draft:** Jenny J. W. Liu, Natalie Ein, Julia Gervasio.

**Writing – review & editing:** Jenny J. W. Liu, Natalie Ein, Julia Gervasio, Bethany Easterbrook, Maede S. Nouri, Anthony Nazarov, J. Don Richardson.

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
