## [Decision Letter · Decision Letter 0]

18 Sep 2023

PONE-D-23-22022Usability and Accuracy of the SWIFT-ActiveScreener Preliminary evaluation for use in clinical researchPLOS ONE

Dear Dr. Liu,

Thank you for submitting your manuscript to PLOS ONE. After careful consideration, we feel that it has merit but does not fully meet PLOS ONE’s publication criteria as it currently stands. Therefore, we invite you to submit a revised version of the manuscript that addresses the points raised during the review process.

We look forward to receiving your revised manuscript.

Kind regards,

Giuseppe Marano

Academic Editor

PLOS ONE

“This work is supported in part by the Atlas Institute for Veterans and Families”

Reviewers' comments:

Reviewer's Responses to Questions

**Comments to the Author**

1. Is the manuscript technically sound, and do the data support the conclusions?

Reviewer #1: Partly

Reviewer #2: Partly

2. Has the statistical analysis been performed appropriately and rigorously? 

Reviewer #1: No

Reviewer #2: Yes

3. Have the authors made all data underlying the findings in their manuscript fully available?

Reviewer #1: Yes

Reviewer #2: Yes

4. Is the manuscript presented in an intelligible fashion and written in standard English?

Reviewer #1: Yes

Reviewer #2: No

5. Review Comments to the Author

Reviewer #1: Overall, I think your study is worth publishing but you need to be careful about the claims you make. This an assessment of a potentially valuable tool that benefits from being independent of the tool developers. It would also be useful if you could extract information about how accurate the tool would be with fewer numbers of screeners, and if you could use a well validated method for assessing usability and usefulness.

Please revise the title of your paper so its clear what Active Screener is.e.g. Usability and Accuracy of the SWIFT-Active Screener tool to support systematic reviews of clinical research

In your Aims section, you justify the study firstly by saying there is little evidence of how Active Screener may perform in evaluations of mental health and treatment outcomes but Howard et al assessed Active Screener on 26 data sets from different fields, what reason is there to believe that mental health field would be different?

The description of the evaluation is not detailed enough for it to be replicated. Some description of how Active Screener works is needed (particularly as it has two facets, i. ordering articles in terms of specified criteria and ii. identifying when a certain percentage of expected relevant papers have been found), and then you need to explain how the 18 participants interacted with the system. Did every screener review each article in the same order, or did screeners who accessed the system at different times see different articles? Did the screeners get any feedback on their performance? You report that your results are better than the results reported by the tool developers, but is this because you have 18 screeners? Would the results be closer to Howard et al.'s results, if you had only two reviewers? In your Aims section you also justify your study by saying " Further, the precision of the estimation model remains unclear", however, Howard et al's estimate of precision was based on the model of the screening process they used in their evaluation. If you found better precision, this could be due to your evaluation process being based on 18 screeners.

For assessing personal opinions you would have better to have used one of the well known technology evaluation methods, for example the Technology Acceptance Model (TAM) which uses a number of questions (similar to those you used) to assess the usability and usefulness of a technology. It is much better to used a previously validated model than invent you own set of likert-format ordinal scales. Also, you should be precise about the nature of questions, you seem to be equating an open question with a free text response as qualitative data, and calling any closed questions quantitative data, even those referring to subjective opinions based on a likert-format agreement scale. Another advantage of the TAM is that their is a process for aggregating the collected data into an overall assessment of the subjective factors. See any of the many papers discussing the model such as:

Marangunić, N., Granić, A. Technology acceptance model: a literature review from 1986 to 2013. Univ Access Inf Soc 14, 81–95 (2015). https://doi.org/10.1007/s10209-014-0348-1.

Richard J. Holden, Ben-Tzion Karsh, The Technology Acceptance Model: Its past and its future in health care,

Journal of Biomedical Informatics, Volume 43, Issue 1, 2010, Pages 159-172,

Reviewer #2: This is an interesting and useful paper. I have outlined my queries and comments below (in the order they appear in the manuscript). The most important of which is point 3. I hope you find them helpful.

1. The flow of the introduction could be improved. For example:

You provide a statement about the aim of the paper in the middle of the introduction (page 3, sentence beginning "In this paper...") and repeat it later on. I suggest removing this.

You introduce ActiveScreener and it's purpose of reducing screening time (page 3, final sentence of 1st paragraph), then discuss the process of conducting reviews and resource issue (page 3, paragraph 2), then introduce software to support review (page 4, paragraph 1). It would make more sense to me discuss the process of conducting reviews and resource issues then introduce software more generally then talk specifically about ActiveScreener.

I would be interested to know why you chose ActiveScreener over other software. Does it offer something different to other tools?

2. The descrition of how the 'accuracy' part of the paper was conducted is not clear to me. For example, what does it mean to set the accuracy of ActiveScreener to 95? Was ActiveScreener trained using the initial 5390 references, then the remaining 4612 references were assessed by both ActiveScreener and human readers? Does assessment happen independently, or do the human readers see ActiveScreener inclusion/exclusion decisions? Perhaps it could be presented as a flow diagram?

3. Throughout the paper you refer to the 'accuracy' of assessment. Accuracy implies that inclusion/exclusion decisions are being compared against the truth (i.e. that a paper should be included in the review because it meets the inclusion criteria at the point of full text assessment). But this isn't what is being assessed with the method employed. It is agreement between ActiveScreener and the human readers on decisions at title/abstract assessment. You could change 'accuracy' to 'agreement' throughout. But this doesn't tell us anything about whether ActiveScreener is categorising papers appropriately (which is what I, as a reviewer, need it to do). You need to compare decisions made at title/abstract to final decisions at full text assessment to do this.

4. Agreement is present between ActiveScreener and the whole cohort of human screeners. It would be useful to have this information by human experience so we can see if the tool is more/less helpful for different levels of experience. Especially as half of the samples has not conducted a review before.

5. Do you haev any extra information about the 97 papers for which there was no agreement between ActiveScreener and the human screeners? Were the human readers correct to include these papers? Was there something different about them that made it hard for ActiveScreener to interpret them?

6. Please can you provide some desciption of how the survey was developed. Has it been assessed for reliability and validity? If yes, what are they? If not, why not? What are the implications for using a survey that has not been assessed for reliability/validity (to add to the discussion, where appropriate)?

7. The discussion states ActiveScreener can reduce screening time and effort (page 17, paragraph). Where in the methods does it explain how amount of screening time and effort are assessed? And where in the results does it reportthe results of these?

6. PLOS authors have the option to publish the peer review history of their article (what does this mean?). If published, this will include your full peer review and any attached files.

Reviewer #1: **Yes: **Barbara A. Kitchenham

Reviewer #2: No

---

## [Author Response · Author response to Decision Letter 0]

12 Oct 2023

Dear Dr. Marano and Reviewers,

We would like to thank you for your appraisal of our manuscript “Usability and Accuracy of the SWIFT-ActiveScreener: Preliminary evaluation for use in clinical research”. We have carefully considered each of the comments provided, and attempted to address each point below. We thank you for giving us the opportunity to revise and improve our manuscript.

Response to Reviewers

Editor Revisions

1. Please ensure that your manuscript meets PLOS ONE's style requirements, including those for file naming. The PLOS ONE style templates can be found at:

Response: We have amended our manuscript to ensure that it follows PLOS ONE’s style requirements provided above.

2. Thank you for stating the following financial disclosure: “This work is supported in part by the Atlas Institute for Veterans and Families”

Response: The funders had no role in this study aside from funding its completion. As such, our financial disclosure statement should be amended to report: “This work is supported in part by the Atlas Institute for Veterans and Families. The funders had no role in study design, data collection and analysis, decision to publish, or preparation of the manuscript.” This has been included in the revised cover letter.

Response: We have addressed the above prompts in our revised cover letter and have uploaded a minimal de-identified dataset.

Reviewer One 

1. Overall, I think your study is worth publishing but you need to be careful about the claims you make. This an assessment of a potentially valuable tool that benefits from being independent of the tool developers. It would also be useful if you could extract information about how accurate the tool would be with fewer numbers of screeners, and if you could use a well validated method for assessing usability and usefulness.

Response: We thank the reviewer for taking the time to provide feedback. We agree that it would be useful if we were able to extract information regarding the accuracy of the tool with fewer numbers of screeners, and hope to assess this as part of our future work. As such, we have added the following sentence: “Future directions of this research should consider the assessment of ActiveScreener AI accuracy when including fewer numbers of screeners. Previous research in other areas indicates that ActiveScreener maintains high levels of accuracy with as few as two reviewers [9] and duplication of this result would be beneficial for use in mental health-related systematic reviews with more limited resources.”

2. Please revise the title of your paper so its clear what Active Screener is.e.g. Usability and Accuracy of the SWIFT-Active Screener tool to support systematic reviews of clinical research

Response: As recommended by the reviewer, we have changed the manuscript title to the following: “Usability and Accuracy of the SWIFT-ActiveScreener Systematic Review Support Tool: Preliminary evaluation for use in clinical research”

3. In your Aims section, you justify the study firstly by saying there is little evidence of how Active Screener may perform in evaluations of mental health and treatment outcomes but Howard et al assessed Active Screener on 26 data sets from different fields, what reason is there to believe that mental health field would be different? 

Response: While Howard at al. did assess ActiveScreener performance using 26 different datasets, these were primarily studies conducted in genomics, toxicology, and animal models, which may lend themselves to more “straightforward” inclusion and exclusion criteria than the mental health field. Due to the inherent challenges for measurement in psychology (please see Osborne 2010 “Challenges for quantitative psychology and measurement in the 21st century”), and that the mental health field continues to update its diagnostic criteria for numerous diagnoses, the creation of inclusion/exclusion criteria, and training reviewers on said criteria, may become more complicated. We felt that due to these challenges it was important to discern whether the ActiveScreener AI was able to accurately predict which articles should and should not be included during the initial phase of screening when conducting a mental-health focused review. We have added more information in the manuscript to clarify this. 

4. The description of the evaluation is not detailed enough for it to be replicated. Some description of how Active Screener works is needed (particularly as it has two facets, i. ordering articles in terms of specified criteria and ii. identifying when a certain percentage of expected relevant papers have been found), and then you need to explain how the 18 participants interacted with the system. Did every screener review each article in the same order, or did screeners who accessed the system at different times see different articles? Did the screeners get any feedback on their performance? You report that your results are better than the results reported by the tool developers, but is this because you have 18 screeners? Would the results be closer to Howard et al.'s results, if you had only two reviewers? In your Aims section you also justify your study by saying " Further, the precision of the estimation model remains unclear", however, Howard et al's estimate of precision was based on the model of the screening process they used in their evaluation. If you found better precision, this could be due to your evaluation process being based on 18 screeners.

Response: We appreciate the feedback that our description was not specific enough, and as such, have cited Howard et al. description of the ActiveScreener software with the following additions: “Once records have been imported into ActiveScreener, the software uses topic modeling, which is the clustering of related documents to discover computationally derived common themes, which are then used in combination with the training set of data [8]. In this case, the “training set” is the real time manual identification of uploaded documents relevant or not relevant. Based on user feedback via patterns of screening, ActiveScreener uses its pretrained algorithm to build a log-linear model estimating the probability that an article should be included versus excluded. This AI prioritization of articles is believed to be most relevant during screening, and can trim screening time and human effort by nearly 70% [8]”

We have further clarified the role of the screeners and how they accessed ActiveScreener on their own time in the Part 1 procedure. This now reads “A total of 10002 references required review at the title and abstract stage. ActiveScreener accuracy statistics were set at 95% resulting in 5390 of these references to be reviewed in duplicate by screeners. Screeners were able to access ActiveScreener at any time on their own schedules, and when they logged on, would be provided with the most relevant article at that time as identified by ActiveScreener rather than every screener reviewing articles in the same order.”

We agree that a higher number of screeners may have partially led to the increased accuracy of the prediction software as suggested by the reviewer. While we believe that the high accuracy of ActiveScreener would have been maintained with a lower number of screeners, this determination was not one of our objectives and as such, we have added the following to the beginning of our Discussion: “While the software was expected to reach 95% accuracy, the actual accuracy of its machine learning model in our review exceeded 95% (97.9%), which may have been aided by the high number of independent screeners on this project.”

5. For assessing personal opinions you would have better to have used one of the well known technology evaluation methods, for example the Technology Acceptance Model (TAM) which uses a number of questions (similar to those you used) to assess the usability and usefulness of a technology. It is much better to used a previously validated model than invent you own set of likert-format ordinal scales. Also, you should be precise about the nature of questions, you seem to be equating an open question with a free text response as qualitative data, and calling any closed questions quantitative data, even those referring to subjective opinions based on a likert-format agreement scale. Another advantage of the TAM is that their is a process for aggregating the collected data into an overall assessment of the subjective factors. See any of the many papers discussing the model such as: Marangunić, N., Granić, A. Technology acceptance model: a literature review from 1986 to 2013. Univ Access Inf Soc 14, 81–95 (2015). https://doi.org/10.1007/s10209-014-0348-1.

Richard J. Holden, Ben-Tzion Karsh, The Technology Acceptance Model: Its past and its future in health care, Journal of Biomedical Informatics, Volume 43, Issue 1, 2010, Pages 159-172

Response: We appreciate the reviewer’s detailed comments regarding the use of an already validated tool rather than designing our own survey to assess usability and experiences with ActiveScreener. While TAM is an excellent measure for acceptability of technologies, we made the decision to create our own survey to assess the explicit experiences with ActiveScreener, as well as the training that was provided to learn how to use this software. The TAM is meant for a wide range of technologies and as such, did not have the degree of specificity that we hoped to obtain in responses (e.g., regarding whether there were glitches in the software). 

We appreciate the reviewer’s comment regarding qualitative vs. quantitative data. We decided to use the term qualitative data to describe the open text boxes, as we thematically coded the responses. As this is an interpretation-based, descriptive analysis of the text responses provided by screeners, we felt it was most in-line with using the term “qualitative” to describe the approach. 

Reviewer Two

This is an interesting and useful paper. I have outlined my queries and comments below (in the order they appear in the manuscript). The most important of which is point 3. I hope you find them helpful.

1. The flow of the introduction could be improved. For example: You provide a statement about the aim of the paper in the middle of the introduction (page 3, sentence beginning "In this paper...") and repeat it later on. I suggest removing this.

Response: We have removed the stated aim from earlier in the introduction as recommended.

2. You introduce ActiveScreener and it's purpose of reducing screening time (page 3, final sentence of 1st paragraph), then discuss the process of conducting reviews and resource issue (page 3, paragraph 2), then introduce software to support review (page 4, paragraph 1). It would make more sense to me discuss the process of conducting reviews and resource issues then introduce software more generally then talk specifically about ActiveScreener. 

Response: We appreciate the reviewer’s comment regarding improving the flow of our paper. As suggested, we have removed the initial introduction to ActiverScreener on page 1. 

3. I would be interested to know why you chose ActiveScreener over other software. Does it offer something different to other tools?

Response: We chose ActiveScreener over other software options to due its ability to classify and predict whether an article should be included, while also providing users with information regarding predictive screening accuracy. This software specifically provides the opportunity to reduce time spent on initial abstract and title screening, which can be arduous depending on how wide or narrow the developed inclusion criteria are. In order to clarify this point, we have the following to the introduction:“In this paper, we evaluated ActiveScreener in terms of its accuracy and usability in a large SR of mental health outcomes following treatment for PTSD. ActiveScreener was selected for this project namely for its departure from programs that uses AI to identify records, and instead, uses machine learning to build a predictive algorithm to reduce time spent in screening phases of SRs.”

4. The description of how the 'accuracy' part of the paper was conducted is not clear to me. For example, what does it mean to set the accuracy of ActiveScreener to 95? Was ActiveScreener trained using the initial 5390 references, then the remaining 4612 references were assessed by both ActiveScreener and human readers? Does assessment happen independently, or do the human readers see ActiveScreener inclusion/exclusion decisions? Perhaps it could be presented as a flow diagram?

Response: Thank you for your comment. The algorithm that ActiveScreener uses was set to 95% accuracy, which meant that in order to reach the threshold of 95% of studies that “should” be included, being included in the full-text screening, 5390 references had to be reviewed by the systematic review screeners. ActiveScreener is trained on an ongoing basis, based on the dataset, and following which types of articles are included/excluded during this initial phase. When the accuracy statistics were reset to 100%, this was the way to tell the ActiveScreener algorithm that every single paper which “should” be included in the review needed to be captured, leading to the screening of the remaining 4612 references. Human screeners do not have access to the individual decisions of ActiveScreener as they are screening, but can see a graph visualization of how “close” the screening has gotten to including 95% of the appropriate articles. In order to clarify how this software works, we have added text at the end of the introduction and in the Part 1 procedure. 

5. Throughout the paper you refer to the 'accuracy' of assessment. Accuracy implies that inclusion/exclusion decisions are being compared against the truth (i.e. that a paper should be included in the review because it meets the inclusion criteria at the point of full text assessment). But this isn't what is being assessed with the method employed. It is agreement between ActiveScreener and the human readers on decisions at title/abstract assessment. You could change 'accuracy' to 'agreement' throughout. But this doesn't tell us anything about whether ActiveScreener is categorising papers appropriately (which is what I, as a reviewer, need it to do). You need to compare decisions made at title/abstract to final decisions at full text assessment to do this. 

Response: Thank you for your comment. We have changed accuracy to agreement where appropriate. Agreement was high between screeners and ActiveScreener for the first phase of screening (ti

---

## [Decision Letter · Decision Letter 1]

20 Nov 2023

PONE-D-23-22022R1Usability and accuracy of the SWIFT-ActiveScreener systematic review support tool:  Preliminary evaluation for use in clinical researchPLOS ONE

Dear Dr. Liu,

Thank you for submitting your manuscript to PLOS ONE. After careful consideration, we feel that it has merit but does not fully meet PLOS ONE’s publication criteria as it currently stands. Therefore, we invite you to submit a revised version of the manuscript that addresses the points raised during the review process.

We look forward to receiving your revised manuscript.

Kind regards,

Giuseppe Marano

Academic Editor

PLOS ONE

Reviewers' comments:

Reviewer's Responses to Questions

**Comments to the Author**

1. If the authors have adequately addressed your comments raised in a previous round of review and you feel that this manuscript is now acceptable for publication, you may indicate that here to bypass the “Comments to the Author” section, enter your conflict of interest statement in the “Confidential to Editor” section, and submit your "Accept" recommendation.

Reviewer #2: (No Response)

Reviewer #3: (No Response)

2. Is the manuscript technically sound, and do the data support the conclusions?

Reviewer #2: No

Reviewer #3: Yes

3. Has the statistical analysis been performed appropriately and rigorously? 

Reviewer #2: Yes

Reviewer #3: Yes

4. Have the authors made all data underlying the findings in their manuscript fully available?

Reviewer #2: Yes

Reviewer #3: No

5. Is the manuscript presented in an intelligible fashion and written in standard English?

Reviewer #2: Yes

Reviewer #3: Yes

6. Review Comments to the Author

Reviewer #2: Thank you for your considered responses and revisions to my review.

My remaining issue is with accuracy/agreement. If the human reviewers and ActiveScreener are both terrible at identifying relevant literature (e.g. they miss 99% of studies that would be included at full text assessment) they might well have excellent ‘agreement’. But they are not ‘accurate’. Agreement with poor performance would introduce more problems than it would solve.

Unfortunately, the study design employed here doesn’t enable us to understand the impact that ActiveScreener would have on the review process. Without being able to compare the title/abstract decisions to the final inclusion/exclusion decisions, the conclusions about the usefulness of ActiveScreener should be tempered. Revisions might include:

1. Make it clear in the methods section that what is being examined is the agreement between ActiveScreener and the reviewers’ decisions at title/abstract screening.

2. Use the term ‘agreement’ throughout.

3. Acknowledge that because the decisions at title/abstract assessment are not compared against final inclusion decisions, it’s not possible to tell what impact ActiveScreener would have on the review process.

Reviewer #3: This article is well-written and it is clear that a lot of work was done to evaluate the tool. Especially the high number of screeners that used the tool contributes to the evaluation's quality. My comments are mostly about the clarity of descriptions of how the tool works, and what the AI predicts.

I do agree with reviewer 1 comment 4, where they mentioned that the description of how ActiveScreener works needs to be more clear. In the updated manuscript you added this sentence: 'the software uses topic modeling, which is the clustering of related documents to discover computationally derived common themes, which are then used in combination with the training set of data [8].' but to the best of my knowledge this statement is incorrect according to the description provided by Howard et al. (2020), who explicitly mention a L2-regularized log-linear model. They do refer to an older paper which describes a different software that, among others, uses topic modelling but they do not mention any topic models as part of ActiveScreener. Howard et al (2020) also do not mentioned that ActiveScreener uses a pre-trained model, they do in fact describe active-learning which means that the model is trained from scratch for every new review project. You are correct to point out that the model is a log-linear model, but this is used to re-order the references as they are screened, so the screeners are always presented with the most likely relevant references. This first algorithm does not predict if an article is relevant or irrelevant, it sorts them by likely relevancy. A potential description for this model could be along the lines of 'ActiveScreener uses a L2-regularized log-linear model and active-learning approach. This means that the AI is continually trained during the screening process; whenever screeners identify included articles the model is re-trained and orders likely relevant references to the top of the screening pile'.

What remains unclear in the revised manuscript is that the system combines two different AI approaches: firstly the active learning (ie. re-ordering of references during screeing with the log-linear model described above) and secondly the statistical algorithm predicting the 95% inclusion rate, as described by Howard et. al (2020). It is this second algorithm that tells screeners when to stop, and thus causes the time-savings. It might be beneficial if this distinction was clear in the manuscript, as previously noted by the other reviewer as well. For the second model, a description could be along the lines of 'Sensitivity within the screening process is then estimated using a statistical model based on a negative binomial distribution. This model alerts screeners when a certain threshold, for example 95% of all likely relevant articles, have been included (Howard et al. 2020)'

This sentence “While the software was expected to reach 95% accuracy, the actual accuracy of its machine learning model in our review exceeded 95% (97.9%), which may have been aided by the high number of independent screeners on this project.” ActiveScreener predicts the inclusion rate of relevant articles, in other words: the sensitivity (and not accuracy). It might be a good idea to replace 'accuracy' with 'sensitivity' in this sentence if I don't misunderstand your evaluation. Sensitivity is the fraction of included references that were found during screening, ie. 95% sensitivity means that 95% of includes were found at the time when the tool told screeners to stop. Or in your case, 97.9% were actually found, which shows that the tool was conservative and its prediction on the safe side, as also described by Howard et. al (2020) with their 26 evaluation reviews. Since you are also evaluating other aspects of the system and the way how the log-linear model performs it might be worth thinking about using the term 'reliability', for example in the title. Could you please consider reviewing the use of the word 'accuracy' in all contexts and consider using 'sensitivity', or 'reliability' or 'agreement' whenever you did not explicitly calculate the metric known as 'accuracy'?

Line 115 please consider using 'ActiveScreener inclusion statistics were set at 95% predicted sensitivity [..]' or 'predicted inclusion rate' for clarity.

It is unclear to me how ActiveScreener predicted a reference as conflicted? Do you mean that there was an unresolved screening conflict between reviewers and that the tool pointed out that there were conflicts?

I do agree with your response about the TAM and that there was indeed value in explicitly evaluating the tool.

7. PLOS authors have the option to publish the peer review history of their article (what does this mean?). If published, this will include your full peer review and any attached files.

Reviewer #2: No

Reviewer #3: No

---

## [Author Response · Author response to Decision Letter 1]

12 Dec 2023

Dear Dr. Marano and Reviewers,

We would like to thank you for your further appraisal of our manuscript titled “Usability and Agreement of the SWIFT-ActiveScreener: Preliminary evaluation for use in clinical research”. We have attempted to address each new point provided by reviewers for consideration below. We thank you for giving us another opportunity to revise and improve our manuscript.

Response to Reviewers

Reviewer Two 

Thank you for your considered responses and revisions to my review.

Response: We appreciate the time the reviewer took to provide feedback to our manuscript, and have attempted to address comments below.

1. My remaining issue is with accuracy/agreement. If the human reviewers and ActiveScreener are both terrible at identifying relevant literature (e.g. they miss 99% of studies that would be included at full text assessment) they might well have excellent ‘agreement’. But they are not ‘accurate’. Agreement with poor performance would introduce more problems than it would solve. Unfortunately, the study design employed here doesn’t enable us to understand the impact that ActiveScreener would have on the review process. Without being able to compare the title/abstract decisions to the final inclusion/exclusion decisions, the conclusions about the usefulness of ActiveScreener should be tempered. 

Response: We thank the reviewer for taking the time to provide further feedback for the improvement of this manuscript. We acknowledge that agreement with poor performance would be problematic and introduce a host of negative issues when attempting to use a tool such as this, and have tried to implement your suggestions throughout. 

2. Make it clear in the methods section that what is being examined is the agreement between ActiveScreener and the reviewers’ decisions at title/abstract screening.

Response: As recommended by the reviewer, included the following sentence in the methods section “Screening results for the remaining 4612 references from ActiveScreener and the screeners were then compared to assess agreement between ActiveScreener and reviewer’s decisions during title/abstract screening.”

3. Use the term ‘agreement’ throughout.

Response: We have attempted to use the term agreement rather than accuracy throughout the manuscript as the request of the reviewer.

4. Acknowledge that because the decisions at title/abstract assessment are not compared against final inclusion decisions, it’s not possible to tell what impact ActiveScreener would have on the review process. 

Response: We have included a sentence in the discussion section acknowledging that the final impact of ActiveScreener is not captured in this paper. This sentence reads: “Importantly, as decisions at the title and abstract phase were not compared directly against final inclusion decisions in this analysis, it is not clear the magnitude of impact ActiveScreener has on the screener process in its entirety. However, one can assume with relative confidence that due to the high agreement in phase one, high levels of agreement would have been maintained.”

Reviewer Three

This article is well-written and it is clear that a lot of work was done to evaluate the tool. Especially the high number of screeners that used the tool contributes to the evaluation's quality. My comments are mostly about the clarity of descriptions of how the tool works, and what the AI predicts.

Response: We would like to thank the reviewer for this comment, and have attempted to address further comments below.

1. I do agree with reviewer 1 comment 4, where they mentioned that the description of how ActiveScreener works needs to be more clear. In the updated manuscript you added this sentence: 'the software uses topic modeling, which is the clustering of related documents to discover computationally derived common themes, which are then used in combination with the training set of data [8].' but to the best of my knowledge this statement is incorrect according to the description provided by Howard et al. (2020), who explicitly mention a L2-regularized log-linear model. They do refer to an older paper which describes a different software that, among others, uses topic modelling but they do not mention any topic models as part of ActiveScreener. Howard et al (2020) also do not mentioned that ActiveScreener uses a pre-trained model, they do in fact describe active-learning which means that the model is trained from scratch for every new review project. You are correct to point out that the model is a log-linear model, but this is used to re-order the references as they are screened, so the screeners are always presented with the most likely relevant references. This first algorithm does not predict if an article is relevant or irrelevant, it sorts them by likely relevancy. A potential description for this model could be along the lines of 'ActiveScreener uses a L2-regularized log-linear model and active-learning approach. This means that the AI is continually trained during the screening process; whenever screeners identify included articles the model is re-trained and orders likely relevant references to the top of the screening pile'.

Response: We appreciate the time the reviewer has taken to address this statement and ensure the accuracy of our ActiveScreener description, as the description that we used was inadvertently out of date. We have incorporated the reviewer’s suggested changes to the description of how ActiveScreener works. This now reads “ActiveScreener uses a L2-regularized log-linear model and active-learning approach to screening, meaning that the model is continually trained during the screening process [8]. In this case, the “training” occurs whenever a screener identifies included articles through the real time manual identification of uploaded documents. Each time an article is identified as “included” the model is re-trained and re-orders the most likely relevant references to be screened next.”

2. What remains unclear in the revised manuscript is that the system combines two different AI approaches: firstly the active learning (ie. re-ordering of references during screeing with the log-linear model described above) and secondly the statistical algorithm predicting the 95% inclusion rate, as described by Howard et. al (2020). It is this second algorithm that tells screeners when to stop, and thus causes the time-savings. It might be beneficial if this distinction was clear in the manuscript, as previously noted by the other reviewer as well. For the second model, a description could be along the lines of 'Sensitivity within the screening process is then estimated using a statistical model based on a negative binomial distribution. This model alerts screeners when a certain threshold, for example 95% of all likely relevant articles, have been included (Howard et al. 2020) 

Response: We agree that it would be beneficial for readers to understand that the system itself uses two separate AI approaches to re-order likely relevant articles and generate sensitivity estimates. We have therefore included the following sentence in the description of the tool, which is based on the reviewer’s feedback: “A statistical model based on a negative binomial distribution is then used by ActiveScreener to estimate sensitivity of the screening process, and is used to alert screeners that a specified threshold of likely relevant articles (in this case, 95%) have been included.”

3. This sentence “While the software was expected to reach 95% accuracy, the actual accuracy of its machine learning model in our review exceeded 95% (97.9%), which may have been aided by the high number of independent screeners on this project.” ActiveScreener predicts the inclusion rate of relevant articles, in other words: the sensitivity (and not accuracy). It might be a good idea to replace 'accuracy' with 'sensitivity' in this sentence if I don't misunderstand your evaluation. Sensitivity is the fraction of included references that were found during screening, ie. 95% sensitivity means that 95% of includes were found at the time when the tool told screeners to stop. Or in your case, 97.9% were actually found, which shows that the tool was conservative and its prediction on the safe side, as also described by Howard et. al (2020) with their 26 evaluation reviews. Since you are also evaluating other aspects of the system and the way how the log-linear model performs it might be worth thinking about using the term 'reliability', for example in the title. Could you please consider reviewing the use of the word 'accuracy' in all contexts and consider using 'sensitivity', or 'reliability' or 'agreement' whenever you did not explicitly calculate the metric known as 'accuracy'?

Response: Throughout the text, we have replaced the term accuracy with either agreement or sensitivity in order to reduce confusion and ensure clarity of our processes. 

4. Line 115 please consider using 'ActiveScreener inclusion statistics were set at 95% predicted sensitivity [..]' or 'predicted inclusion rate' for clarity. 

Response: We have added the term “predicted inclusion rate” in order to improve clarity of this sentence as requested.

5. It is unclear to me how ActiveScreener predicted a reference as conflicted? Do you mean that there was an unresolved screening conflict between reviewers and that the tool pointed out that there were conflicts? 

Response: ActiveScreener only identified a reference as conflicted if two human screeners disagreed on whether the reference should be included or excluded. When we requested reviewers to stop screening in order to assess agreement, there were unresolved conflicts which were captured as conflicted. As such, the confusion matrix predictive agreement included a class for conflicted references. This was done to see if ActiveScreener would be able to predict with good agreement, articles which would be difficult for human screeners to decide whether to include or exclude. In this case, we found that the human screeners included 97 articles which the confusion matrix had identified as conflicted. 

6. I do agree with your response about the TAM and that there was indeed value in explicitly evaluating the tool.

Response: Thank you for your comment. We will be sure to consider the use of the TAM in future research where appropriate, as it is a useful evaluation tool.

---

## [Decision Letter · Decision Letter 2]

11 Jan 2024

PONE-D-23-22022R2

Usability and agreement of the SWIFT-ActiveScreener systematic review support tool Preliminary evaluation for use in clinical research

PLOS ONE

Dear Dr. Liu,

Thank you for submitting your manuscript to PLOS ONE. After careful consideration, we have decided that your manuscript does not meet our criteria for publication and must therefore be rejected.

I am sorry that we cannot be more positive on this occasion, but hope that you appreciate the reasons for this decision.

Kind regards,

Giuseppe Marano

Academic Editor

PLOS ONE

Reviewers' comments:

Reviewer's Responses to Questions

**Comments to the Author**

1. If the authors have adequately addressed your comments raised in a previous round of review and you feel that this manuscript is now acceptable for publication, you may indicate that here to bypass the “Comments to the Author” section, enter your conflict of interest statement in the “Confidential to Editor” section, and submit your "Accept" recommendation.

Reviewer #2: (No Response)

Reviewer #3: All comments have been addressed

2. Is the manuscript technically sound, and do the data support the conclusions?

Reviewer #2: No

Reviewer #3: Yes

3. Has the statistical analysis been performed appropriately and rigorously? 

Reviewer #2: Yes

Reviewer #3: Yes

4. Have the authors made all data underlying the findings in their manuscript fully available?

Reviewer #2: Yes

Reviewer #3: No

5. Is the manuscript presented in an intelligible fashion and written in standard English?

Reviewer #2: Yes

Reviewer #3: Yes

6. Review Comments to the Author

Reviewer #2: Thank you for your helpful revisions. A couple of final remarks.

1. There are a few places where it would be helpful to be more explicit about what is being done by adding “at the title/abstract assessment phase of the review” at the end of each sentence, i.e.

• Line 40 - inclusion or exclusion of relevant articles “at the title/abstract assessment phase of the review”.

• Line 107 - Eighteen screeners were trained to identify articles for inclusion and exclusion “at the title/abstract assessment phase of the review”.

• Line 239 - In our study, we found that ActiveScreener performed above its expected 95% agreement in prediction “at the title/abstract assessment phase of the review”

2. There’s a sentence in the abstract and conclusion (lines 43 and 309) that I don’t think accurately reflects the state of knowledge regarding ActiveScreener:

• Our results showed that ActiveScreener, when used appropriately, may save considerable time and human resources when performing SR

In the absence of evidence about whether the title/abstract decisions were correct, we cannot tell what impact ActiveScreener will have on the review process. We only know that it performed similarly to the sample of human reviewers at this stage of the review.

Reviewer #3: Thank you for addressing the comments in detail, the paper looks good now. I recommended it for publication.

7. PLOS authors have the option to publish the peer review history of their article (what does this mean?). If published, this will include your full peer review and any attached files.

Reviewer #2: No

Reviewer #3: No

- - - - -

---

## [Author Response · Author response to Decision Letter 2]

8 Feb 2024

Reviewer #2:

Thank you for your helpful revisions. A couple of final remarks.

Response: We thank the reviewer for taking the time to provide further feedback for the improvement of this manuscript.

1. There are a few places where it would be helpful to be more explicit about what is being done by adding “at the title/abstract assessment phase of the review” at the end of each sentence, i.e.

• Line 40 - inclusion or exclusion of relevant articles “at the title/abstract assessment phase of the review”.

• Line 107 - Eighteen screeners were trained to identify articles for inclusion and exclusion “at the title/abstract assessment phase of the review”.

• Line 239 - In our study, we found that ActiveScreener performed above its expected 95% agreement in prediction “at the title/abstract assessment phase of the review”

Response: We have added the requested phrases into the manuscript as suggested.

2. There’s a sentence in the abstract and conclusion (lines 43 and 309) that I don’t think accurately reflects the state of knowledge regarding ActiveScreener:

• Our results showed that ActiveScreener, when used appropriately, may save considerable time and human resources when performing SR

Response: We appreciate the reviewer’s comment and have adjusted these sentences accordingly.

In the absence of evidence about whether the title/abstract decisions were correct, we cannot tell what impact ActiveScreener will have on the review process. We only know that it performed similarly to the sample of human reviewers at this stage of the review.

Response: Thank you for your comment. We have noted this as a limitation within our discussion section.

Reviewer #3: 

Thank you for addressing the comments in detail, the paper looks good now. I recommended it for publication.

Response: Thank you for dedicating your time to review our manuscript. Your insightful feedback was immensely appreciated and will undoubtedly contribute to enhancing the quality and impact of this manuscript.

---

## [Decision Letter · Decision Letter 3]

8 May 2024

PONE-D-23-22022R3Usability and agreement of the SWIFT-ActiveScreener systematic review support tool Preliminary evaluation for use in clinical researchPLOS ONE

Dear Dr. Liu,

Thank you for submitting your manuscript to PLOS ONE. After careful consideration, we feel that it has merit but does not fully meet PLOS ONE’s publication criteria as it currently stands. Therefore, we invite you to submit a revised version of the manuscript that addresses the points raised during the review process.

Please revise the sections in the article that express inaccurate meanings, such as English grammar, spelling, sentence structure, etc.

Please add some figures to improve readability. Please make this revision before considering this study for publication.

We look forward to receiving your revised manuscript.

Kind regards,

Guanghui Liu

Academic Editor

PLOS ONE

Reviewers' comments:

Reviewer's Responses to Questions

**Comments to the Author**

1. If the authors have adequately addressed your comments raised in a previous round of review and you feel that this manuscript is now acceptable for publication, you may indicate that here to bypass the “Comments to the Author” section, enter your conflict of interest statement in the “Confidential to Editor” section, and submit your "Accept" recommendation.

Reviewer #3: All comments have been addressed

Reviewer #4: (No Response)

Reviewer #5: All comments have been addressed

2. Is the manuscript technically sound, and do the data support the conclusions?

Reviewer #3: Yes

Reviewer #4: Partly

Reviewer #5: No

3. Has the statistical analysis been performed appropriately and rigorously? 

Reviewer #3: Yes

Reviewer #4: Yes

Reviewer #5: Yes

4. Have the authors made all data underlying the findings in their manuscript fully available?

Reviewer #3: Yes

Reviewer #4: Yes

Reviewer #5: Yes

5. Is the manuscript presented in an intelligible fashion and written in standard English?

Reviewer #3: Yes

Reviewer #4: Yes

Reviewer #5: Yes

6. Review Comments to the Author

Reviewer #3: Thank you for the revision, I believe this article is ready for publication with only one minor change; but I believe the authors can address this without the need of further review on my end.

I am sorry for missing this tiny detail in my previous review. You revised the description of ActiveScreener's machine-learning according to my previous comments but I just realised one likely incorrect sentence remained: page 4 line 84 about the pretrained model. ActiveScreener uses active learning, and NOT a pretrained model. This was described by Howard et. al in your cited reference in section 2.2, bag-of-words and tf-idf is used to represent text and then a L2-regularized log-linear model and active-learning follows. As someone who works with these models on a daily basis, I can tell you that there is an important difference between pretrained and tf-idf initialised models.

I believe there are two ways forward: could you please delete the sentence about the pretrained model that isn't supported by the cited literature; or if you would like to keep it, could you please contact Sciome and confirm that a pretrained model is actually used? If you do decide to contact them, I found their email address: swift-activescreener@sciome.com

Reviewer #4: The authors assessed the agreement and usability of the ActiveScreener in the large SR on mental health outcomes following treatment for PTSD. However, there are several issues that need to be addressed before this study can be considered for publication.

1) In the 'Introduction' section, the authors should focus on reviewing current methods for conducting SRs and in particular compare their disadvantages and advantages.

2) There are so many sections that it becomes confusing and unreadable, the authors should re-integrate the whole article, e.g. integrate the methods and results of Part I and Part II separately.

3) Authors need to pay attention to the formatting of the article, pauses between paragraphs, spaces at the beginning of paragraphs, etc.

4) What is innovative about the article? What are the advantages and disadvantages of ActiveScreener compared to other tools?

5) This manuscript requires careful editing and special attention to English grammar, spelling and sentence structure.

Reviewer #5: Thank you for addressing the comments in detail, The paper looks good. I recommended it for publication.

7. PLOS authors have the option to publish the peer review history of their article (what does this mean?). If published, this will include your full peer review and any attached files.

Reviewer #3: No

Reviewer #4: **Yes: **Xiaodong Zou

Reviewer #5: No

---

## [Author Response · Author response to Decision Letter 3]

24 May 2024

Thank you for reconsidering our manuscript titled "Usability and agreement of the SWIFT-ActiveScreener systematic review support tool Preliminary evaluation for use in clinical research". We have thoroughly addressed all remaining comments and suggestions below and in the manuscript.

Response to Reviewers

Reviewer #3: 

I am sorry for missing this tiny detail in my previous review. You revised the description of ActiveScreener's machine-learning according to my previous comments but I just realised one likely incorrect sentence remained: page 4 line 84 about the pretrained model. ActiveScreener uses active learning, and NOT a pretrained model. This was described by Howard et. al in your cited reference in section 2.2, bag-of-words and tf-idf is used to represent text and then a L2-regularized log-linear model and active-learning follows. As someone who works with these models on a daily basis, I can tell you that there is an important difference between pretrained and tf-idf initialised models.

I believe there are two ways forward: could you please delete the sentence about the pretrained model that isn't supported by the cited literature; or if you would like to keep it, could you please contact Sciome and confirm that a pretrained model is actually used? If you do decide to contact them, I found their email address: swift-activescreener@sciome.com

Response: We appreciate your attention to detail in order to improve our paper. As requested, we have deleted the sentence (lines 88-89).

Reviewer #4: (response required)

The authors assessed the agreement and usability of the ActiveScreener in the large SR on mental health outcomes following treatment for PTSD. However, there are several issues that need to be addressed before this study can be considered for publication.

In the 'Introduction' section, the authors should focus on reviewing current methods for conducting SRs and in particular compare their disadvantages and advantages.

Response: We appreciate the reviewer’s feedback and have provided references throughout the introduction to protocols which provide “gold standard” systematic review methodologies for readers to refer to when conducting a systematic review. We have included the following sentences: “Standardized protocols, such as the Preferred Reporting Items for Systematic Reviews and Meta-Analysis (PRISMA), offer researchers a roadmap to conducting SRs with rigour and fidelity [3]. In addition, formal guides established by Cochrane provide further evaluation criteria in order to provide appropriate context for the interpretation of study data in various research settings [4].” (lines 54-58)

Further, we have included the typical methodology for screening articles for inclusion while conducting a systematic review on lines 64-70. As the software we reviewed primarily reduced the workload during the screening process, we felt it was prudent to focus on this component of a systematic review rather than reviewing the entire SR process, which is outside the scope of this manuscript. 

There are so many sections that it becomes confusing and unreadable, the authors should re-integrate the whole article, e.g. integrate the methods and results of Part I and Part II separately.

Response: We have integrated the methods and results as requested by the reviewer. 

Authors need to pay attention to the formatting of the article, pauses between paragraphs, spaces at the beginning of paragraphs, etc.

Response: Thank you for your recommendation. We have thoroughly reviewed and corrected the formatting issues, ensuring proper pauses between paragraphs and consistent spacing at the beginning of each paragraph. We believe these changes improve the overall readability and presentation of the article.

What is innovative about the article? What are the advantages and disadvantages of ActiveScreener compared to other tools?

Response: This innovation of this article is due to the fact that it is the first manuscript to assess the utility of ActiveScreener using a large scale, mental health focused systematic review. Over ten thousand references required review during the title and abstract screening phase of this project, leading to a large number of articles from which to compare agreement between human screeners and the ActiveScreener algorithm. Other previous articles which have reviewed ActiveScreener’s utility have assessed it using smaller numbers of articles and different research areas. 

For additional reader clarification, we have also included information regarding the novelty of ActiveScreener compared to other, currently used systematic review tools such as Covidence on lines 77-86, which read: “For example, COVIDENCE aids clinical research reviews with its ability to distinguish between randomized controlled trials (RCTs) versus non-RCTs. Other systematic review tools include similar options to apply sorting tags and take notes. Critically, ActiveScreener is a novel SR tool due to its use of active learning, which allows the software to estimate completeness of the screening process, and notifies reviewers when they may stop screening early. In this paper, we evaluated ActiveScreener in terms of its agreement with human screeners, and usability in a large SR of mental health outcomes following treatment for PTSD. ActiveScreener was selected for this project namely for its departure from programs that use AI to identify records, and instead, use machine learning to build a predictive algorithm to reduce time spent in screening phases of SRs.” 

This manuscript requires careful editing and special attention to English grammar, spelling and sentence structure.

Response: We appreciate your feedback and have made adjustments to grammar, spelling, and sentence structure throughout the manuscript in consideration of this concern. 

Reviewer #5: 

Thank you for addressing the comments in detail, The paper looks good. I recommended it for publication.

Response: Thank you for your positive feedback and recommendation for publication. We're thrilled to hear that the revisions have met your expectations and look forward to sharing our work with a wider audience.

---

## [Decision Letter · Decision Letter 4]

6 Sep 2024

PONE-D-23-22022R4Usability and agreement of the SWIFT-ActiveScreener systematic review support tool: Preliminary evaluation for use in clinical researchPLOS ONE

Dear Dr. Liu,

Thank you for submitting your manuscript to PLOS ONE. After careful consideration, we feel that it has merit but does not fully meet PLOS ONE’s publication criteria as it currently stands. Therefore, we invite you to submit a revised version of the manuscript that addresses the points raised during the review process.

I recommend that the authors have their manuscript reviewed by a native English speaker. Additionally, it is crucial that they carefully follow the reviewers’ recommendations.

We look forward to receiving your revised manuscript.

Kind regards,

Giuseppe Marano

Academic Editor

PLOS ONE

Journal Requirements:

Additional Editor Comments (if provided):

Reviewers' comments:

Reviewer's Responses to Questions

**Comments to the Author**

1. If the authors have adequately addressed your comments raised in a previous round of review and you feel that this manuscript is now acceptable for publication, you may indicate that here to bypass the “Comments to the Author” section, enter your conflict of interest statement in the “Confidential to Editor” section, and submit your "Accept" recommendation.

Reviewer #6: All comments have been addressed

Reviewer #7: (No Response)

2. Is the manuscript technically sound, and do the data support the conclusions?

Reviewer #6: Yes

Reviewer #7: Yes

3. Has the statistical analysis been performed appropriately and rigorously? 

Reviewer #6: Yes

Reviewer #7: I Don't Know

4. Have the authors made all data underlying the findings in their manuscript fully available?

Reviewer #6: Yes

Reviewer #7: Yes

5. Is the manuscript presented in an intelligible fashion and written in standard English?

Reviewer #6: Yes

Reviewer #7: Yes

6. Review Comments to the Author

Reviewer #6: I have no further review comments to authorship team at this point. This has undergone extensive peer review already

Reviewer #7: Thank you for the opportunity to review this manuscript. I am happy to see more literature on the effectiveness of machine learning to increase screening efficiency in the title and abstract phase of systematic reviews (SRs), given the importance of not missing relevant records when performing this step.

I have a couple of follow-up questions pertaining to question 4 above (full availability of underlying data) and about the ethics statement response provided with the manuscript, which indicated that an ethics statement was not applicable: I may have missed it, but I could not see the supporting information files with the underlying data. However, I admit to not being entirely familiar with the particular requirements for the granularity of the data and I answered "yes" based on the statement provided by the authors.

I was also wondering if it would be pertinent to explain why ethics review was not needed, given the involvement of human participants in the survey component in particular. Would this be considered a quality improvement project?

There are a few small comments I have regarding some of the information provided in the manuscript.

Line 56: PRISMA is a reporting guideline rather than a guide to conducting SRs: I would suggest adjusting the wording accordingly, per reference 3 cited, which states "PRISMA 2020 is not intended to guide systematic review conduct, for which comprehensive resources are available." See also https://link.springer.com/article/10.1186/s13643-021-01671-z for an editorial on this point.

Line 79-80: I would suggest rewording this phrase a little, e.g., to "ActiveScreener is part of a growing set of novel tools that use active learning..." - Covidence includes this functionality since 2022 (see https://www.covidence.org/blog/release-notes-december-2022-machine-learning/) and other products have been using active learning for several years now as well (DistillerSR AI, EPPI-Reviewr to name a couple). If ActiveScreener is novel in a particular way, it would be worth emphasizing how (for example, are there other screening tools using active learning that notify reviewers when they may stop screening? Is it novel in that particular way?)

I found the manuscript to be clear and effective at communicating the results.

Thanks again for providing this information about ActiveScreener, which will hopefully soon be available for other researchers to use to support their own decisions about how to undertake the screening process more efficiently and feasibly.

7. PLOS authors have the option to publish the peer review history of their article (what does this mean?). If published, this will include your full peer review and any attached files.

Reviewer #6: **Yes: **Ronald Chow

Reviewer #7: No

---

## [Author Response · Author response to Decision Letter 4]

13 Sep 2024

Response to Reviewers

Reviewer #6 

I have no further review comments to authorship team at this point. This has undergone extensive peer review already

● Author Response: Thank you for your feedback and for confirming that no further comments are needed at this stage. We appreciate the time and effort dedicated to the extensive peer review process and are pleased that the revisions have met the necessary standards.

Reviewer #7 

Thank you for the opportunity to review this manuscript. I am happy to see more literature on the effectiveness of machine learning to increase screening efficiency in the title and abstract phase of systematic reviews (SRs), given the importance of not missing relevant records when performing this step. I have a couple of follow-up questions pertaining to question 4 above (full availability of underlying data) and about the ethics statement response provided with the manuscript, which indicated that an ethics statement was not applicable: I may have missed it, but I could not see the supporting information files with the underlying data. However, I admit to not being entirely familiar with the particular requirements for the granularity of the data and I answered "yes" based on the statement provided by the authors.

● Author Response: The authors would like to thank the reviewer for their time and efforts providing a comprehensive review of this paper. To clarify the question raised with respect to the availability of underlying data, we included all available raw data in Table 1 (ActiveScreener data) and Table 2 (survey data) within the manuscript. 

I was also wondering if it would be pertinent to explain why ethics review was not needed, given the involvement of human participants in the survey component in particular. Would this be considered a quality improvement project?

● Author Response: Yes, because this survey was administered as a normal operational requirement for quality assurance and improvement, this survey would be considered a quality improvement project and would not require REB review per the Tri-Council Interagency Advisory Panel on Research Ethics (TCPS 2, 2022; Chapter 2, Article 2.5). 

There are a few small comments I have regarding some of the information provided in the manuscript. Line 56: PRISMA is a reporting guideline rather than a guide to conducting SRs: I would suggest adjusting the wording accordingly, per reference 3 cited, which states "PRISMA 2020 is not intended to guide systematic review conduct, for which comprehensive resources are available." See also https://link.springer.com/article/10.1186/s13643-021-01671-z for an editorial on this point.

● Author Response: We have adjusted the wording to better reflect the nature of the reporting guidelines (see highlighted p.3, ll. 54-57)

Line 79-80: I would suggest rewording this phrase a little, e.g., to "ActiveScreener is part of a growing set of novel tools that use active learning..." - Covidence includes this functionality since 2022 (see https://www.covidence.org/blog/release-notes-december-2022-machine-learning/) and other products have been using active learning for several years now as well (DistillerSR AI, EPPI-Reviewr to name a couple). If ActiveScreener is novel in a particular way, it would be worth emphasizing how (for example, are there other screening tools using active learning that notify reviewers when they may stop screening? Is it novel in that particular way?)

● Author Response: We agree and made the appropriate edits (see highlighted p.4, ll. 83-86)

I found the manuscript to be clear and effective at communicating the results.

● Author Response: Thank you for this comment.

Thanks again for providing this information about ActiveScreener, which will hopefully soon be available for other researchers to use to support their own decisions about how to undertake the screening process more efficiently and feasibly.

● Author Response: Thank you for this comment.

---

## [Decision Letter · Decision Letter 5]

10 Oct 2024

Usability and agreement of the SWIFT-ActiveScreener systematic review support tool: Preliminary evaluation for use in clinical research

PONE-D-23-22022R5

Dear Dr. Liu,

We’re pleased to inform you that your manuscript has been judged scientifically suitable for publication and will be formally accepted for publication once it meets all outstanding technical requirements.

Kind regards,

Giuseppe Marano

Academic Editor

PLOS ONE

Additional Editor Comments (optional):

Reviewers' comments:

Reviewer's Responses to Questions

**Comments to the Author**

1. If the authors have adequately addressed your comments raised in a previous round of review and you feel that this manuscript is now acceptable for publication, you may indicate that here to bypass the “Comments to the Author” section, enter your conflict of interest statement in the “Confidential to Editor” section, and submit your "Accept" recommendation.

Reviewer #6: All comments have been addressed

Reviewer #7: All comments have been addressed

2. Is the manuscript technically sound, and do the data support the conclusions?

Reviewer #6: Yes

Reviewer #7: Yes

3. Has the statistical analysis been performed appropriately and rigorously? 

Reviewer #6: N/A

Reviewer #7: I Don't Know

4. Have the authors made all data underlying the findings in their manuscript fully available?

Reviewer #6: Yes

Reviewer #7: Yes

5. Is the manuscript presented in an intelligible fashion and written in standard English?

Reviewer #6: Yes

Reviewer #7: Yes

6. Review Comments to the Author

Reviewer #6: Manuscript reads well, and appreciate the authorship team in their efforts through the peer-review process. I have no further comments.

Reviewer #7: (No Response)

7. PLOS authors have the option to publish the peer review history of their article (what does this mean?). If published, this will include your full peer review and any attached files.

Reviewer #6: **Yes: **Ronald Chow

Reviewer #7: No

---

## [Editor Report · Acceptance letter]

11 Oct 2024

PONE-D-23-22022R5 

PLOS ONE

Dear Dr. Liu, 

I'm pleased to inform you that your manuscript has been deemed suitable for publication in PLOS ONE. Congratulations! Your manuscript is now being handed over to our production team.

Kind regards, 

on behalf of

Dr. Giuseppe Marano 

Academic Editor

PLOS ONE